# Enduring Effects of the COVID-19 Pandemic on Food Access, Nutrition, and Well-Being in Rural Appalachia

**DOI:** 10.3390/ijerph21050594

**Published:** 2024-05-04

**Authors:** Kathryn M. Cardarelli, Emily DeWitt, Rachel Gillespie, Nathan Bandy, Heather Norman-Burgdolf

**Affiliations:** 1Department of Health, Behavior & Society, University of Kentucky, Lexington, KY 40506, USA; rachel.gillespie@uky.edu; 2Department of Dietetics and Human Nutrition, University of Kentucky, Lexington, KY 40506, USA; emily.dewitt@uky.edu (E.D.); nathan.bandy@uky.edu (N.B.); heather.norman@uky.edu (H.N.-B.)

**Keywords:** food access, food security, rural health, Appalachia, health disparities, nutrition, health equity, mental health

## Abstract

The COVID-19 pandemic produced acute effects on health inequities, yet more enduring impacts in vulnerable populations in rural Appalachia are understudied. This qualitative study included three focus groups with thirty-nine adults (74% female, mean age 52.7 years) to obtain perspectives on the impact of the COVID-19 pandemic on well-being in Martin County, Kentucky, in fall 2022. Grounded Theory was employed using an iterative inductive-deductive approach to capture the lasting effects of the COVID-19 pandemic on health practices and status. Three prominent themes emerged: (1) increased social isolation; (2) household cost of living strains caused by inflation; and (3) higher food prices and diminished food availability causing shifts in food purchasing and consumption. Participants noted that the rising cost of living resulted in residents having to “choose between medication, food and utilities”. Increased food prices resulted in residents “stretching” their food, modifying how they grocery shopped, and limiting meat consumption. Persistent food shortages were exacerbated by there being few grocery stores in the county. Lastly, increased social isolation was profoundly articulated as widely impacting mental health, especially among youth. Our findings underscore the ongoing deleterious effects of inflation and food supply chain disruptions in this rural, geographically isolated community, which resulted in difficult spending choices for residents.

## 1. Introduction

The COVID-19 pandemic has exacerbated existing health inequities in the United States (U.S.), with individuals of lower socioeconomic positions experiencing greater severe illness and mortality [1]. These individuals are also more susceptible to poor food access, defined as limited availability of foods and a decreased ability to afford foods in their local community [2]. Furthermore, the pandemic caused supply disruptions to the food chain, impeding food access, and rural communities with limited grocery stores have been particularly vulnerable to this challenge [3]. To date, few studies have explored the lasting impact of the pandemic on food access and food insecurity in rural U.S. communities. However, studies exploring the acute phase of the pandemic (2020–2021) in diverse communities around the U.S. noted there was an increase in food insecurity prevalence and food-related worries. In rural Vermont, those experiencing job loss were 3 times more likely to experience food insecurity. Notably, those for whom an increase in food insecurity was observed because of COVID-19 were newly food insecure [4]. In New Mexico, food-insecure individuals reported difficulty procuring food and had to travel to more than one location to source food for their household. This was compounded by worries about food becoming more expensive and fear of not being able to afford food [5]. In New York, those experiencing food-related worries and negative job impact since the start of the COVID-19 pandemic were more likely to report food insecurity compared to those without these challenges [6]. Studies also suggest women and households with children were more likely to be food insecure [4]. Like other states across the U.S., Kentucky was negatively impacted by the COVID-19 pandemic [3]. Certain areas of the state already experience challenges to food access and food insecurity, notably, the eastern part of the state, situated in Appalachia. The Appalachian region of the U.S. spans across 13 states from the southern part of New York to the northern region of Mississippi. Additionally, rural Appalachia is disproportionately susceptible to food insecurity and poor food access due to numerous systemic and infrastructure challenges. Appalachian communities may be uniquely impacted by the lasting effects of the COVID-19 pandemic due to persistent poverty, limited healthcare access, and geographic isolation [7]. For decades, Appalachian populations have experienced higher chronic disease prevalence, and Appalachian Kentucky has some of the highest rates of obesity, diabetes, heart disease, and cancer in the nation [7], with food insecurity in the region identified as one of the significant contributors [7]. Nearly one-quarter of adults in the Central Appalachian region report living with a disability [7], and residents of Central Appalachia report 42% more physically unhealthy days per month and 25% higher mentally unhealthy days per month than the average U.S. adult [7]. The COVID-19 pandemic may have further impacted these outcomes, making it important to understand how to improve well-being in rural Appalachia.

Supply chain disruptions and inflation resulted in higher food prices during the acute phase of the COVID-19 pandemic, and the impact of these factors on food security status in vulnerable populations has been documented by our team and others [3,4]. However, little research exists to document the potential persistent, more enduring effects of these factors on mental and physical well-being in vulnerable rural populations. Rural, geographically isolated communities such as those in Appalachia have fewer food security assets [7], and public health professionals seeking to boost food security in these populations have a need for more recent findings. Now, several years past the acute phase of the pandemic, our team sought to fill this gap in the evidence. Considering food prices are a key determinant of food insecurity [8], we explored how food access and prices may have impacted food purchasing and dietary consumption habits, as well as overall well-being, in one rural Appalachian community. Specifically, we sought to understand the implications of the COVID-19 pandemic on food access and sourcing, nutrition-related behaviors, and overall health status in 2022 to inform health promotion interventions in this population. We employed a qualitative approach to obtain a deep understanding of the perspectives and insights of this target group in order to inform culturally acceptable interventions.

## 2. Materials and Methods

### 2.1. Setting and Design

Focus groups were conducted in Martin County, Kentucky, in October 2022, focusing on the enduring implications of the COVID-19 pandemic on health behaviors such as diet and physical activity. This work is part of a larger, multi-year High Obesity Program (HOP) project funded by the Centers for Disease Control and Prevention (CDC) to reduce rural obesity prevalence [9,10]. Martin County, located in eastern Kentucky along the border of West Virginia, has an adult obesity prevalence greater than 40%, as well as a high prevalence of cancer and other obesity-related chronic illnesses [7]. This community faces persistent poverty, with a median household income of $29,386 and an estimated 40.5% of individuals living in poverty [11], and one in five adults is considered food insecure [12]. The food environment in this community offers few nutritious food options with only two grocery stores within county lines, and many residents travel out of the county to shop for groceries in supercenters if they have transportation access [13]. Residents of Martin County experience characteristics of poor quality of life, with nearly 25% of adults reporting poor or fair health and more than five poor physical and mental health days per month [14]. Given the social and economic conditions, Martin County is designated as “highly vulnerable” according to the CDC’s Social Vulnerability Index [15].

We chose to employ a qualitative approach in this study for several reasons. First, qualitative data can be powerful when capturing the lived experiences of individuals experiencing health inequities, such as our study population, and these experiences can often not be sufficiently characterized by quantitative data alone [16]. Second, because health inequities are deeply rooted in complex social, cultural, economic, and political contexts, qualitative approaches enable us to explore these nuances and contextual factors that shape well-being. Lastly, a qualitative approach to data collection provides a platform for health disparity populations to share their stories and insights, which can inform more culturally acceptable interventions [17].

### 2.2. Sample

Adults from Martin County were recruited in fall 2022 to participate in focus groups. This non-probability sample was recruited with assistance from the Martin County Wellness Coalition and Martin County Cooperative Extension Office, both of whom disseminated study recruitment materials in person at community events and in the Extension Office, which is viewed as a trusted source of community information. In remote rural areas such as our target community, it can be challenging to obtain a comprehensive sampling frame or to randomly select participants due to logistical difficulties and limited resources. Additionally, Facebook is a critical source of communication in the county, and recruitment materials were shared via the Martin County Cooperative Extension Office Facebook page to promote the event to community members. Participants were eligible to participate if they were at least 21 years old, currently resided in Martin County, had been a resident of the county for at least one year, and had the ability to be physically active. All participants received a $40 (USD) gift card for their time and participation.

### 2.3. Measures

Prior to partaking in conversations, all participants reviewed and signed written informed consent and completed a brief demographic survey that captured age, race/ethnicity, gender, highest educational or technical training attainment, income, employment status, household size, and nutrition assistance use.

Food security status was captured using the validated six-item food security module from the United States Department of Agriculture [18]. Participants were asked about the amount of food consumed and the ability to afford food within the last 12 months. All participant responses were scored using the validated scoring procedure associated with the USDA screener, and affirmative responses were tallied to calculate food security status. Those with 0–1 affirmative responses were classified as having high or marginal food security; 2–4 low food security; 5–6 very low food security. Those with scores representing low or very low food security were considered food insecure.

A semi-structured moderator guide (located in Appendix A) was developed among the study team to guide focus group conversations. Open-ended questions aimed to assess a range of responses related to the impacts of the COVID-19 pandemic regarding food decisions, including food access and food security in the community and the current rising food costs, physical activity engagement and access, and other possible pandemic-related behaviors and health conditions. Each focus group was facilitated by a trained moderator (K.M.C) with three team members taking notes and monitoring digital audio recording devices (N.B, E.D., R.G). Data saturation was achieved after three focus group conversations. Each conversation lasted approximately one hour, and all were held at the Martin County Extension Office. All study procedures and materials were approved by the University of Kentucky Institutional Review Board (protocol #40895).

### 2.4. Analysis

Each focus group was audio-recorded on multiple devices and transcribed verbatim via the transcription service. A Grounded Theory approach [19] was utilized as three investigators (K.M.C, R.G., N.B.) reviewed the transcripts. This approach was selected in order to allow the theory to emerge from the data, rather than imposing an a priori theme or hypothesis on the data [20,21]. Themes in the data were identified using an iterative inductive-deductive approach to capture the lasting effects of the COVID-19 pandemic on food choice and physical activity. The codebook was created through an iterative and flexible process that emerged through open, focused, axial, and theoretical coding of data, along with constant comparison and refinement. The three investigators independently coded the same data and met to discuss and resolve any discrepancies in their coding, reaching a consensus after multiple rounds of coding on the final codes and categories [22,23,24]. Final themes were agreed upon by all co-authors.

## 3. Results

### 3.1. Sample Characteristics

Thirty-nine adults participated in three focus groups (see Table 1). The sample was predominantly female (n = 29, 74%) and white (n = 38, 97%). Participants ranged in age from 21 to 87 years (52.7 ± 18.52 SD years), with almost half being retired (n = 17, 45%), though nearly as many were currently employed (n = 14). Nearly three-fourths of the sample reported a household size of 2–4 people (n = 28), which was similar to the mean household size for the county of 3.1 persons. An annual household income of less than $50,000 (USD) was most frequently reported (n = 28, 72%), though many participants had some college education or were college graduates (n = 14, 36% and n = 11, 28%, respectively). Among participants, 61.5% (n = 24) reported high or marginal food security, which is lower than the county food security estimate of approximately 80% [12]. Of our study participants reporting food insecurity, 40% (n = 6) experienced low food security and 60% (n = 9) experienced very low food security. With regard to receiving nutrition assistance, 15% of participants reported receiving Supplemental Nutrition Assistance Program (SNAP) benefits, and 8% reported receiving Women, Infants and Children (WIC) benefits. Another 8% reported receiving nutrition assistance from food pantries. Of those participants using food assistance programs, 1/3 reported still experiencing food insecurity.

### 3.2. Qualitative Findings

Three emergent themes were revealed from the focus group conversations: (1) persistent social isolation, which resulted in increased mental health challenges, (2) household cost of living strains caused by inflation, and (3) diminished food availability/higher food prices, causing shifts in food purchasing and consumption patterns. These themes are described below, and additional illustrative quotes from participants are provided in Table 2.

#### 3.2.1. Persistent Social Isolation

Although the focus group questions did not focus on mental health, participants spoke at great length about how pandemic-related social isolation resulted in increased mental illness. Many participants discussed growing accustomed to being inside their homes because of the COVID-19 pandemic risk guidance and how this culture resulted in persistent increased social isolation. Participants described this as a result of increased at-home employment and well as attending church via Facebook. One participant noted that these changed modalities resulted in less physical activity: “I think that with those at-home jobs, they’re putting people away from the community, so they don’t walk or do any activity or anything like that anymore”. Participants also overwhelmingly connected this to an increase in the prevalence of mental health challenges, including depression. “Mental health has suffered greatly”. “Everybody feels isolated”. Of note about the impact on youth mental health, one participant shared, “You got a lot more bullying in the schools. You get a lot more outbursts in the schools because they’ve been home, and now they’re in a more structured environment and it’s causing their anxiety to go through the roof”.

#### 3.2.2. Household Cost of Living Strains Caused by Inflation

The high level of poverty (40.5%, [11]) experienced by Martin County residents, coupled with the economic inflation that occurred in 2022, resulted in significant financial strain on households. Residents reported spikes in the cost of gasoline, rent, utilities, car taxes, and groceries (see next theme for details on food costs) over the previous year. Because many families live on fixed incomes in this community, participants shared that, because of economic inflation, families were having to make difficult decisions about spending priorities, including, for example, foregoing expenses for medications in order to pay rent or utility bills. Many participants spoke about the lack of well-paying jobs in the county and that the impact of inflation was significant on households in their community. One participant suggested that the county was “recession-proof” because of ongoing economic challenges: “I say that is because it’s constantly in a depressive state kind of thing”.

#### 3.2.3. Higher Food Prices and Diminished Food Availability Causing Shifts in Food Purchasing and Consumption

Participants spoke at great length about the increased cost of food, especially meat and eggs. Because of these increased prices, participants described efforts to make their food “stretch”, modifying how they grocery shopped, cooking without it (e.g., substituting “soup beans” for meat), purchasing food less often, looking to hunting to fill freezers, and raising chickens as strategies to cope. One participant noted, “My husband’s a meat eater, and I’ve had to cut him back”. Another remarked, “Now before, I’d make vegetable soup. I’d put a whole pound of ground beef in it. Now you put that in half, you put half a pound in it and save the other half for another meal. You just have to stretch what you can get”. “I cannot afford meat, so I’ve had to rearrange my menus, my breakfast, suppers from what am I going to make now kind of thing. And what I can afford to buy, purchase, other than meat”.

In every focus group, participants described diminished availability of food items in grocery stores. They discussed the disruptions that occurred to the food supply and speculated on contributing reasons, including labor shortages (e.g., “lots of the trucks ain’t getting to the stores” due to lack of truck drivers) and cargo shipping operations not being able to fully operate. Participants described the disruption of food supply chains as not just impacting individuals and families, but also the local school system’s ability to source food items. “At my position in my work [within the local school system], we’ve had to make adjustments to school meals and their main entrees”. Further challenging the availability of food was the lack of farmers participating in the local farmers’ market and the lack of grocery stores in the community.

As a consequence of these conditions, residents reported shifting their purchasing to less nutritious, less expensive food. As reported previously [13], this rural community has only two grocery stores in the county, and while many residents expressed a desire to shop locally, increased food prices were repeatedly mentioned as a barrier. Dollar Stores have expanded in the region, and one participant noted, “Dollar Stores have stepped up and they got more stuff, but it’s still not enough variety”. Several participants cited the need to weigh the cost of gas to travel out of the county for groceries against the high prices in their local grocery store.

## 4. Discussion

While the initial focus of this study was to understand the impact of the pandemic on food access and nutrition, participants shared experiences beyond physical health that reflected the far-reaching effects of the COVID-19 pandemic on mental health and overall well-being in the Appalachian region. The participant experiences highlight the potential lasting implications of the COVID-19 pandemic in one rural, Appalachian community. Participants not only described the enduring effects of high inflation, increased food costs, and diminished food availability, but also described the lasting consequences of persistent social isolation on their mental health.

Social isolation and mental health quickly emerged in discussions related to food access, physical activity, and overall health status. As previously noted, adults in Central Appalachia disproportionately experience disability and poor mental and physical health compared to the average U.S. adult [7]. The social isolation described by our study population as an enduring effect of the pandemic is alarming, given the existing chronic health challenges of the region. Moreover, the challenges posed by social isolation are of growing concern in rural communities where familial support, friendship, and places of community gathering and/or worship, such as churches, are essential for supporting strong social networks [25,26,27]. Our findings are consistent with other work exploring the impact of the acute phase of the COVID-19 pandemic on mental health in Appalachia, in which isolation, loneliness, depression, and anxiety were commonly reported [28,29,30]. Observational data at the national level reflect rises in anxiety and depression during the early period of the pandemic, with economically vulnerable populations experiencing more mental health challenges compared to other groups [31]. However, there is little exploration of the lingering effects of the pandemic on mental health and overall well-being. This highlights the need for holistic approaches supporting mental and physical health that consider individual and social components when establishing health-promoting interventions for rural Appalachian residents.

Our findings indicate residents in this impoverished community struggled financially due to inflation, which resulted in challenges in procuring food. These findings are similar to what was observed in New Mexico during the acute phase of the pandemic when sourcing food was difficult [5]. This may have been compounded by the challenge of living on a fixed income, for individuals who are retired or rely on federal financial assistance. That is, residents in the county were less able to absorb increased costs and were forced to make difficult household decisions on spending. This can further exacerbate food insecurity among limited-resource populations. Participants in the present study described multiple tactics to cope with this challenge and made changes to food purchasing patterns, including stretching their food supplies, eating less nutritious foods, and cutting out meat. While consuming certain meat products less often could result in health improvements, the pervasive strategy of purchasing less expensive and less nutritious foods could counterbalance this. In other work assessing perceptions of food shortages among U.S. adults during the COVID-19 pandemic, those shopping in-store were more likely to anticipate food shortages, and meat was most often expected to be unavailable compared to other foods [32]. It is also unknown from our sample whether cutting out meat was entirely due to high costs, or a shortage driven by supply chain issues. Further, altered food purchasing patterns may have been compounded by the closure of one of the three county grocery stores in early 2020 and the farmers’ market in summer 2021, resulting in a consistent decline in food-sourcing venues during the acute phase of the pandemic [13]. Thus, locals experienced diminished food access and availability requiring them to alter their food procurement patterns. Participants of the current study reported often traveling further to grocery shops, due to limited local options, and perceived higher prices, and they spoke about including the cost of gasoline when gauging grocery shopping options.

Our participants reported lower food insecurity than population estimates for this community reported by Feeding America [12]. This gap may be explained by the timing of our focus groups occurring after increased federal assistance and stimulus checks tapered in Kentucky. As has been previously reported [33], the federal expansion of certain nutritional programs, such as Supplemental Nutrition Assistance Program (SNAP) and the National School Lunch Program operating in the public school systems, provided temporary alleviation of food insecurity by increasing benefit dollars and operationalizing daily mobile meal route distributions throughout communities. However, these strategies are limited to those who are eligible and/or enrolled in these programs, which could highlight the systemic and infrastructural barriers that contribute to pervasive food insecurity, as SNAP has demonstrated poor uptake among eligible populations due to certain enrollment barriers [33,34]. Previous work in this community demonstrated additional COVID-19 nutrition assistance supporting food sufficiency [3,27], yet current findings demonstrate the deleterious effects of the loss of these resources on food decision-making. In light of the recent end of certain federal initiatives that helped mitigate the hardship of the pandemic, such as the Child Tax Credit and SNAP Emergency Allotments, many low-income individuals are now facing a looming hunger crisis once again [35]. Policymakers should consider the behavioral response, such as what was reported in our study, on the broader health and economic landscape when assessing the potential implications of such policy changes on food and nutrition support programs on a national scale. However, future research is needed to determine the longitudinal impacts of fluctuating federal support programs on persistent food insecurity and overall health outcomes.

As noted in the present study, inflation and fixed income influence food choice, and decision-making is harder when fewer options are available. With the closure of local food outlets, members of this community have been forced to shift their food purchasing to other venues. Dollar Stores were noted as a key food-sourcing venue in Martin County, and this finding is notable, given the poverty level of the community. Since 2008, Dollar Stores have been the fastest-growing retail outlet for food procurement and the share of food purchased at dollar stores increases as household income decreases, particularly in rural areas [36]. These venues may also influence the consumption of foods that are higher in calories and lower in nutrients [37]. When transportation is limited, shopping at Dollar Stores may be the only viable solution for residents. Resources highlighting nutritious food options available at these venues could be instrumental in helping individuals and families stretch their food dollars while simultaneously supporting more balanced diets to support health, such as what has been done in other nontraditional food outlets in rural communities already [38]. Ensuring these resources and initiatives are community-specific is critical, and although our study highlights the perspectives of individuals in one specific Appalachian community, these findings may be insightful for other rural highly impoverished or isolated communities, as the widespread impacts of the pandemic are still being understood.

Although this community faces numerous challenges, churches and faith-based organizations were mentioned often as important assets. Belonging to a church is a norm in this rural community, and churches are an important organizing resource that provides care and services for residents, including food, transportation, and childcare. Churches have been effective partners for other areas of wellness, including nutrition and physical activity interventions [39,40], and may be valuable collaborators in supporting health promotion. Beyond health-promoting partnerships, churches serve as a key venue for social connectedness. This is notable in rural communities, as religious organizations commonly provide weekly and seasonal life rhythms in addition to structuring social connections and relationships [41,42]. Previously, churches were identified as key locations of support in this community during the COVID-19 pandemic, including non-traditional food access points, and for their role in social connection regularly [25]. As observed in the present study, the limited ability to engage with church congregations during the COVID pandemic had ripple effects on participant perceptions of mental health. However, the potential role of churches in building resilience to external shocks such as the pandemic is unclear.

Our study has several limitations. Our sample was purposively recruited and therefore may introduce a degree of selection bias. Specifically, our sample was not representative of the county based on two variables: age and educational attainment. The older and more educated adults in our sample may suggest that the challenges described here are underreported relative to the county. Participant responses may have been influenced by social desirability bias since conversations were facilitated by study personnel in the presence of others. Additional inquiry should continue to monitor these impacts and recognize that vulnerable communities remain at risk for additional challenges. Future studies are necessary to better understand how the economic and social disruptions created by the COVID-19 pandemic may exacerbate health inequities and nutritional status in socially isolated and disadvantaged communities. For example, with focus group participants sharing that they reduced meat consumption and other higher-cost food products, such as dairy, protein intake should be considered in communities with high rates of chronic disease.

## 5. Conclusions

This study showed the enduring effects of the COVID-19 pandemic in one rural Appalachian community in fall 2022. Our findings demonstrate that, beyond food access, there are economic and mental health challenges that persist, and multiple areas of well-being are impacted by these persistent effects. These results provide potential opportunities for future interventions that are informed by the community. Notwithstanding the immense challenges that this community faced, as one participant noted, “Martin County is very much about family and neighbor and helping one another”. The noted resilience in the community and the value of supporting one another in this rural, Appalachian population will contribute to its endurance.

## Figures and Tables

**Table 1 ijerph-21-00594-t001:** Sociodemographic characteristics of focus group participants (n = 39) and of Martin Country, Kentucky, Residents, 2022.

Characteristic	Among All Participants *n* (%)	Martin County, KY Residents ^5^ (%)
Age (median)	55 years	39 years
** Gender **		
Female	29 (74%)	45%
Male	10 (26%)	55%
** Race **		
White	38 97%)	92%
** Education ^4^ **		
11th grade and below	2 (5%)	26%
High school graduate or GED	9 (23%)	39%
Some college	14 (36%)	25%
Technical/Trade/Vocational School	3 (8%)	
College graduate	11 (28%)	10%
** Household Income (USD) ^1,2^ **		
<$20,000	10 (26%)	
$20,001–$49,999	18 (47%)	** Median Household Income **
>$50,000	10 (26%)	$29,387
** Employment Status ^1,2^ **		
Employed	14 (37%)	
Unemployed	7 (18%)	
Retired	17 (45%)	

^1^ Some participants chose not to respond. ^2^ No analogous data categories are available from the U.S. Census Bureau. ^4^ Two participants selected some college/trade school; they opted for some college, since they had a higher level of education. ^5^ Data from the U.S. Census Bureau. Abbreviations: KY: Kentucky; GED: Tests of Graduate Educational Development; USD: United States Dollars.

**Table 2 ijerph-21-00594-t002:** Themes and illustrative quotes from focus group participants (n = 39), Martin County, Kentucky, 2022.

Participant Narratives *
Persistent social isolation
“When I go out now, I make a trip count, I don’t like to go in stores and stay more than 10 min. We’re just trying to avoid getting sick”. “Yeah, there’s a lot more depression. There’s anxiety. I think people got used to staying in during COVID and now they can get out, but they’re not getting out and I think that’s causing depression to be a lot worse. And this has affected the younger generation a lot because they were told they couldn’t go to school, they couldn’t be around their friends, and then when they go back to school, they don’t know how to interact with other people anymore”. “No, you can drive by and see that places that churches with plumb full, they’re lucky to have 20 cars, maybe, get that”. “I think COVID left some of us…emotionally traumatized…to where their lifestyle changed dramatically, and there’s been a lot of depression they’re dealing with… We are emotionally, I mean we have been injured basically with COVID”. “COVID made it [depression] more relevant. I guess you could say more people were working from home, more people weren’t seeing family, more people were isolated and that increased it”.
**Household cost of living strains caused by inflation**
“And there’s a lot of people who are very fixed income. So, when the prices of our food changes, a lot of people are going hungry, the utilities are going up. They got to choose between medication, food, and utilities”. “Inflation is deeply affecting our county because the price of gas, the price of groceries. Do I buy groceries, do I pay gas? I got to go to work. We have some people that are trapped”. “And rent is sky high. Most of the people here, they are on disability or fixed incomes and the price of everything keeps going up, and it exceeds what they make. And that’s a big hardship for this county because the prices increase along with big cities and other places that have more job opportunities, the way we make money. But here we don’t have anything basically but minimum wage jobs and so it makes everybody struggle”. “Everything’s so high here in the county, all your bills, water and your sewer bill, I pay both and it’s more than what people know it is in big counties”. “I think our water and sewer is probably top 3% expensive in the whole state. And it’s not even usable, it’s really not even potable water and it’s terrible, it’s just crazy”. “Some of us are on fixed incomes. They may be comfortable fixed incomes, but when everything keeps going up, it’s getting a little difficult”. “Water bill has gone up again. And of course the prices of food, car taxes... it is just getting hard”.
**Higher food prices and diminished food availability causing shifts in food purchasing and consumption**
“Meat is expensive … I mean, you get something for two people that’s big now, and you can’t afford three packages. So, we’ve had to [cut meat], and both of us have lost weight, so that’s a good thing”. “You got to buy stuff that you can actually make meals out of. They can’t have everything they see no…it isn’t just the main entrees any longer that are the priciest items. A lot of times it’s the sides”. “I know the prices at most places, and I end up going to Walmart [outside the county] and I hate that because I’ve always been about spending my money locally. And so, okay, when this first started, the only lunch meet I like is ham and cheese loaf and it was $4.89, save a lot. It was $2 something at Walmart. I mean, you can’t not do that, and you know that you’re going to get stuff cheaper”. “Now before, I’d make vegetable soup, I’d put a whole pound of ground beef in it. Now, you put that in half, you put a pound in it and you save the other half for another meal. You just have to stretch what you can get”. “There’ll be times that we will figure out what we’re going to eat, and sometimes the meal won’t contain meat because both paychecks are just kind of tight. I don’t eat meat—can’t afford it”. “Whereas before I’d keep a gallon of milk…in the fridge at all times. Now I kind of buy the smaller one and not always have milk”. “There’s a lot of empty shelves [in grocery stores]”. “The gasoline diesel prices have skyrocketed…that causes the cost of delivery and foods to get hauled to the markets to go up”.

* Selected illustrative quotes included in table.

## Data Availability

The data in this study may be available by request to the corresponding author.

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
