# Peer review of "Enduring Effects of the COVID-19 Pandemic on Food Access, Nutrition, and Well-Being in Rural Appalachia"

_ijerph, 2024, doi:10.3390/ijerph21050594_

Round 1

Reviewer 1 Report

Comments and Suggestions for Authors

The paper examines the long-term impacts of the COVID-19 pandemic on food access, nutrition and well-being in rural Appalachia. The paper highlights how the pandemic has exacerbated social isolation, increased living costs due to inflation, and led to higher food prices and reduced food availability, significantly affecting residents' purchasing and consumption behaviors.

Criticisms:

1. It is taken for granted that there has been a change in nutrition during COVID; moreover, inflation has affected food spending.The paper offers no innovative insights in this regard. The reduction in meat consumption should be seen as a positive aspect

2..Sample representation does not fully reflect the diversity of the community, limiting the generalizability of results.

3. There is a lack of methodological transparency, especially on participant selection criteria and focus group data analysis.

4. There is a lack of quantitative data to measure the severity of impacts over time.

5. The paper suggests potential interventions but does not evaluate the effectiveness of existing strategies or propose new evidence-based solutions.

It would be helpful to offer more concrete recommendations for policy makers and community organizers, and to suggest specific areas for future research.

Author Response

Reviewer Comment

Author Responses/Revisions

It is taken for granted that there has been a change in nutrition during COVID; moreover, inflation has affected food spending. The paper offers no innovative insights in this regard.

Thank you for underscoring that we had not made clear the innovation of our study. We added details in the Introduction section to clarify that our study is among the first qualitative investigations examining the effect of the covid pandemic on food access/nutrition/well being in this Appalachian population beyond the acute phase (2020-2021) of the pandemic, which is important for public health professionals seeking to mitigate food insecurity in this or similar populations.

The reduction in meat consumption should be seen as a positive aspect.

We agree that it is possible that the noted reduction of meat consumption may translate into positive health effects- this will need additional monitoring. However, we noted in the Discussion section that, “While consuming certain meat products less often could result in health improvements, the pervasive strategy of purchasing less expensive and less nutritious foods could counterbalance this.”

Sample representation does not fully reflect the diversity of the community, limiting the generalizability of results.

In the Discussion section, we underscore the potential selection bias that may have been introduced into our sample based on two characteristics (age and education) that were representative of our study county and the potential impact that may have on our findings.

There is a lack of methodological transparency, especially on participant selection criteria and focus group data analysis.

With regard to participant selection criteria concerns, we added details in the Methods section to explain why a probability sample in this remote rural population is challenging (and why, therefore, we employed a non-probability sample). We also added details to explain how our community partners assisted in recruiting.

With regard to concerns regarding data analysis, we have added details regarding our qualitative data analysis, including justification of the approach, how the codebook was created and information on interrater reliability.

There is a lack of quantitative data to measure the severity of impacts over time.

We recognize the limitations of the quantitative data available to discuss the longitudinal impacts of the pandemic. However, we have attempted to incorporate additional examples in the Introduction and Discussion sections of how food insecurity in particular has changed over a brief period time. We highlight in the discussion areas for future research, how certain policies could have influenced food security status, and the potential implications on long term health outcomes and federal programs.

The paper suggests potential interventions but does not evaluate the effectiveness of existing strategies or propose new evidence-based solutions.

As we state in the Introduction section, the purpose of this manuscript is to fill the current gap in the literature on the potential lasting effects of the COVID-19 pandemic on food insecurity and well-being in a rural Appalachian population. These findings will inform the development of interventions for this population by our team (or other similar populations by other investigators). Therefore, we did not evaluate an existing or evidence-based intervention. Our team will develop future manuscripts that evaluate the effectiveness of the intervention(s).

It would be helpful to offer more concrete recommendations for policy makers and community organizers, and to suggest specific areas for future research.

Thank you for including this recommendation – we have added additional language in the Discussion section that specifically points to areas of priority for future research and policy recommendations.

Reviewer 2 Report

Comments and Suggestions for Authors

Introduction--I appreciated the brevity of the introduction and thought it was sufficient.

Analysis--why use a grounded theory approach? That seems out of place and not appropriate. Please either explain why this approach was taken or revise this. Please describe codebook creation more fully and report interrater reliability. 

Overall it was a well written paper and I appreciated the use of a table to provide illustrative examples of the content. 

Author Response

Reviewer Comment

Author Responses/Revisions

Introduction--I appreciated the brevity of the introduction and thought it was sufficient.

Overall it was a well written paper and I appreciated the use of a table to provide illustrative examples of the content. 

Thank you for these affirming comments.

Analysis--why use a grounded theory approach? That seems out of place and not appropriate. Please either explain why this approach was taken or revise this. Please describe codebook creation more fully and report interrater reliability. 

We used a grounded theory approach to allow the theory to emerge from the data itself, rather than imposing a pre-existing theoretical framework or hypothesis on the data. In our Methods section, we added details on the justification of this approach as well as information on codebook creation and interrater reliability on the final themes.

Reviewer 3 Report

Comments and Suggestions for Authors

The authors examine an important issue. The paper is generally well-written and interesting to read. With that said, more work is needed to prepare this paper for publication.

First, the introduction covers a lot of ground, but it could be reorganized to first start with the issue in broader terms before setting up the focus on food access in Appalachia. As it is written now, it is not clear why the study is needed or what gap if fills.

Second, and related to the first issue, the literature review is very underdeveloped. Before going into detail on the paper's methods and findings, the authors need to expand the lit review to define food access and the food system (as there are multiple definitions in the literature), explain what researchers have written on access in previous studies and more deeply analyze the impacts of covid on food systems. As it stands now, the underdevelopment of the literature review is a significant issue in the paper. The list of references is short and speaks to this shortcoming.

Third, the context of the study could be enhanced. There is no map here to show the reader where the study is situated.  Also, more background could be given on the study context and its food system.

Fourth, the methods are a start but remain underdeveloped. What is the broader approach here in terms of the types of generalizations possible/created and the methods or tools used? More detail is needed on the study selection process and the strengths & weaknesses of using such a method.

Fifth, as the findings are shown in the subsequent sections, the graphics do not speak for themselves. Rather, the author should explain the findings (what they show) and why they are important for the study and broader literature. In this way the findings should be described (what did they show) and analyzed (why are they important).

Sixth, the end of the paper needs to more adequately connect the findings to the broader literature. Since the literature is underdeveloped at the beginning of the paper, it is difficult for the authors to come full circle to explain the important of the findings and the study more broadly. 

Other:

-More graphics would help this paper: map of context, visual flow diagram to show what was done in the paper, and more succinct or readable to table to showcase results

-Title is a bit long, consider condensing

Comments on the Quality of English Language

Writing is generally strong. A read through to improve writing clarity would enhance the paper even more.

Author Response

Reviewer Comment

Author Responses/Revisions

First, the introduction covers a lot of ground, but it could be reorganized to first start with the issue in broader terms before setting up the focus on food access in Appalachia. As it is written now, it is not clear why the study is needed or what gap if fills.

Thank you for underscoring that we had not made clear the innovation of our study. We added details in the introduction to clarify that our study is among the first qualitative investigations examining the effect of the covid pandemic on food access/nutrition/well being in this Appalachian population beyond the acute phase (2020-2021) of the pandemic, which is important for public health professionals seeking to mitigate food insecurity in this or similar populations.

Second, and related to the first issue, the literature review is very underdeveloped. Before going into detail on the paper's methods and findings, the authors need to expand the lit review to define food access and the food system (as there are multiple definitions in the literature), explain what researchers have written on access in previous studies and more deeply analyze the impacts of covid on food systems. As it stands now, the underdevelopment of the literature review is a significant issue in the paper. The list of references is short and speaks to this shortcoming.

Thank you for this comment. We have specifically revised the first paragraph of the Introduction section to include a more robust explanation of food access, and we have incorporated additional references that better analyze the acute impacts of the COVID-19 pandemic on food systems in a variety of ways. While most studies exploring acute effects of food insecurity were centered in other countries or more diverse US locations, we have identified content that is relevant for our study population and the context of rural, Appalachian food insecurity.

Other reviewer feedback specifically addressed the brevity, cohesiveness, and sufficiency of our introduction for this paper. We have addressed your comment while also keeping in mind what other reviewers for this manuscript shared.

Third, the context of the study could be enhanced. There is no map here to show the reader where the study is situated.  Also, more background could be given on the study context and its food system.

We appreciated you noting this geographic element and agree with the request for additional context. We have created and added a map (Figure 1) which denotes where the county of focus is located and precisely where it is situated in the broader United States to enhance readers’ situational perspective.

To support the logical flow of the manuscript, we have included information that provides context for the study setting in the Methods section in 2.1 Setting. Several statistics are provided that provide context regarding the food system within Martin County.

What is the broader approach here in terms of the types of generalizations possible/created and the methods or tools used?

Thank you for this question. While the population living in Appalachian Kentucky has unique characteristics and culture, our findings may be generalizable to other geographically isolated, rural populations. We added the following comment in the Discussion section, “although our study highlights the perspectives of individuals in one specific Appalachian community, these findings may be insightful for other rural highly impoverished or isolated communities, as the widespread impacts of the pandemic are still being understood.”

More detail is needed on the study selection process and the strengths & weaknesses of using such a method.

We added details in the Methods section to explain why a probability sample in this remote rural population is challenging (and why, therefore, we employed a non-probability sample). We also added details to explain how our community partners assisted in recruiting. Additionally, in the Discussion section, we underscore the potential selection bias that may have been introduced into our sample based on two characteristics (age and education) that were representative of our study county and the potential impact that may have on our findings.

Fifth, as the findings are shown in the subsequent sections, the graphics do not speak for themselves. Rather, the author should explain the findings (what they show) and why they are important for the study and broader literature. In this way the findings should be described (what did they show) and analyzed (why are they important).

Thank you for this opportunity to review our manuscript more in depth. We included two tables to summarize our findings. Both tables are referenced in-text within the Results section and explained in narrative form. We have taken the opportunity to better describe what was observed and analyze why these findings are important within the discussion. With additional clarifying information, we would be happy to address this concern further.

Sixth, the end of the paper needs to more adequately connect the findings to the broader literature. Since the literature is underdeveloped at the beginning of the paper, it is difficult for the authors to come full circle to explain the important of the findings and the study more broadly. 

We agree it seems like there is a disconnect between our Discussion and Introduction sections. This is mostly because our findings expanded beyond food access as the only issue impacting this community since the pandemic's acute phase.  While our intent was to assess the impact of COVID-19 on food access and health, our findings ultimately revealed the broader implications of physical and mental health on this rural population.

We have rearranged some of our Discussion section to draw attention to this and to more clearly describe the impact of our findings on broader literature. We have incorporated additional references that connect our findings to the Introduction and the limited literature that is available examining food insecurity in the US during the two years of the pandemic (acute phase).

Title is a bit long, consider condensing.

We have condensed the title- thank you for this suggestion.

Round 2

Reviewer 1 Report

Comments and Suggestions for Authors

The authors have made some improvements to the paper. However, one serious shortcoming remains: there is no numerical data on the responses in this paper and no statistical analysis of them. In a scientific journal, this is too great a limitation. 

Author Response

Reviewer: There is no numerical data on the responses in this paper and no statistical analysis of them.

Response: Our interpretation of this comment is that the reviewer believes the methods are flawed because we did not collect/analyze quantitative data. Setting aside the collection of quantitative data for the characterization of sociodemographic data (see Table 1), we chose to employ a qualitative approach to data collection for this study for a number of reasons. First, qualitative data can be powerful when capturing the lived experiences of individuals experiencing health inequities, such as our study population. These experiences can often not be sufficiently characterized by quantitative data alone. Second, health inequities are deeply rooted in complex social, cultural, economic and political contexts, and qualitative approaches enable us to explore these nuances and contextual factors that shape well being. Lastly, a qualitative approach to data collection provides a platform for health disparity populations to share their stories and insights, which can inform more culturally acceptable interventions. This was our intention. We have added verbiage in the methods section to justify this approach. 

Reviewer 3 Report

Comments and Suggestions for Authors

The authors responded to the reviewers' comments and the paper is stronger for this reason. 

Comments on the Quality of English Language

Generally good. An additional read through to maximize writing quality would be ideal.

Author Response

We reviewed the manuscript for readability and made minor updates. Thank you for your previous suggestions, as we agree that they strengthened the paper.